# Role of Lung Ultrasound in Predicting Clinical Severity and Fatality in COVID-19 Pneumonia

**DOI:** 10.3390/jpm11080757

**Published:** 2021-07-30

**Authors:** Ivan Skopljanac, Mirela Pavicic Ivelja, Ognjen Barcot, Ivan Brdar, Kresimir Dolic, Ozren Polasek, Mislav Radic

**Affiliations:** 1Department of Pulmology, University Hospital of Split, 21000 Split, Croatia; 2Department of Infectious Diseases, University Hospital of Split, 21000 Split, Croatia; mmarendic@gmail.com; 3Department of Surgery, University Hospital of Split, 21000 Split, Croatia; ognjen.barcot@gmail.com; 4Department of Internal Emergency Medicine, University Hospital of Split, 21000 Split, Croatia; ivan_brdar@yahoo.com; 5Department of Diagnostic and Interventional Radiology, University Hospital of Split, 21000 Split, Croatia; kdolic@kbsplit.hr; 6School of Medicine, University of Split, 21000 Split, Croatia; opolasek@gmail.com; 7Division of Rheumatology and Clinical Immunology, Centre of Excellence for Systemic Sclerosis in Croatia, University Hospital of Split, 21000 Split, Croatia; mislavradic@gmail.com

**Keywords:** lung ultrasound, LUS, COVID-19, prognostic, pneumonia

## Abstract

Background: Lung ultrasound (LUS) is a useful imaging method for identifying COVID-19 pneumonia. The aim of this study was to explore the role of LUS in predicting the severity of the disease and fatality in patients with COVID-19. Methods: This was a single-center, follow-up study, conducted from 1 November 2020, to 22 March 2021. The LUS protocol was based on the assessment of 14 lung zones with a total score up to 42, which was compared to the disease severity and fatality. Results: A total of 133 patients with COVID-19 pneumonia confirmed by RT-PCR were enrolled, with a median time from hospital admission to lung ultrasound of one day. The LUS score was correlated with clinical severity at hospital admission (Spearman’s rho 0.40, 95% CI 0.24 to 0.53, *p* < 0.001). Patients with higher LUS scores were experiencing greater disease severity; a high flow nasal cannula had an odds ratio of 1.43 (5% CI 1.17–1.74) in patients with LUS score > 29; the same score also predicted the need for mechanical ventilation (1.25, [1.07–1.48]). An LUS score > 30 (1.41 [1.18–1.68]) and age over 68 (1.26 [1.11–1.43]) were significant predictors of fatality. Conclusions: LUS at hospital admission is shown to have a high predictive power of the severity and fatality of COVID-19 pneumonia.

## 1. Introduction

The coronavirus disease 2019 (COVID-19) is caused by a new strain of the virus discovered at the end of 2019 in China, which had not been detected in humans previously; the World Health Organization named it SARS-CoV-2, and it is the cause of a pandemic that continues to this day [1]. The symptoms of COVID-19 differ between individuals, varying from asymptomatic infection to severe respiratory failure. Common symptoms are a fever, cough, fatigue, slight dyspnea, sore throat, and headache. The vast majority of patients with symptoms and more severe clinical features had one or more comorbidities, for instance, obesity and cardiovascular conditions, with high case fatalities amongst elderly and frail patients [2,3].

Recognizing risk factors at admission which predict disease progression would help physicians to deliver appropriate and timely therapeutic interventions. Increasing age, medical comorbidities, lymphopenia, elevated serum ferritin, d-dimer, cardiac troponin I, C-reactive protein, lactate dehydrogenase, and IL-6 levels are associated with severe illness, poor prognosis, and increased mortality [4,5,6].

COVID-19 pneumonia is radiologically characterized by bilateral pulmonary infiltrates with a propensity toward the lung periphery and a lack of associated pulmonary nodules, cavitation, adenopathy, or pleural effusions [7].

CT is a gold standard for COVID-19 pneumonia imaging, and recent studies have suggested a potential role of CT severity scores in predicting outcomes for SARS-CoV-2 patients [8,9]. However, because CT scans are not readily available in pandemic situations, there is a need for a comparable imaging method. Lung ultrasound (LUS) is a useful noninvasive diagnostic procedure for identifying pleural and pulmonary lesions; it can be performed continuously at the bedside and with no radiation exposure, to assess disease progression and severity. LUS has been shown to be comparable to X-ray and CT scans in the diagnosis of pneumothorax, pleural effusion, and pneumonia, and identifying the signs of cardiac failure [10,11,12,13]. Recent studies have reported that findings on LUS associated with COVID-19 correlate well with CT scans [14,15,16]. In addition, current research suggests the usability of lung ultrasounds for COVID-19 screening at hospital admission, showing a high negative predictive value for LUS [17,18].

Treatments for COVID-19 are mainly in the form of a certain degree of respiratory support with other adjunctive therapies. Supplemental oxygen is the primary step for addressing respiratory impairment, ranging from a low flow nasal cannula, and as the disease progresses, to the usage of masks with higher oxygen delivery, high-flow nasal cannula (HFNC) devices, noninvasive ventilation, and ultimately, to invasive mechanical ventilation [1]. Therefore, it is crucial to be able to predict the degree of oxygen support that patients presenting with COVID-19 pneumonia may need, enabling physicians to adjust other treatment modalities accordingly, such as corticosteroid therapy, or transferring patients to more specialized centers.

There have been some suggestions lately that LUS can predict outcomes in COVID-19; endpoints in most of studies were mortality and the need for invasive mechanical ventilation [19,20,21,22,23,24]. At the beginning of the COVID-19 pandemic, it was observed that LUS did not correlate much with oxygen need in patients at the time of admission to a hospital ward. For example, the patient could be on a low oxygen flow and clinically stable, although the LUS findings were disproportionally negative. Only after a few days would the patient’s clinical status deteriorate significantly, a realization that alerted us to the possible prognostic value of bedside lung ultrasound; therefore, this study explores the role of LUS in predicting the disease severity and mortality in patients with COVID-19.

## 2. Materials and Methods

### 2.1. Study Design

This was a single-center, follow-up study.

### 2.2. Inclusion and Exclusion Criteria

The study was based on a consecutive group of patients admitted with a diagnosis of COVID-19 pneumonia in the University Hospital Centre Split, Croatia, from 1 November 2020, to 22 March 2021. Inclusion criteria were the WHO diagnostic criteria for COVID-19 pneumonia, with SARS-CoV-2 infection confirmed by PCR or a rapid antigen test from a nasopharyngeal swab [25]. Exclusion criteria were pulmonary edema associated with heart failure; severe lung emphysema; chronic interstitial lung disease, severe hemodynamic instability and inability to change body position; severe chest deformity; extensive subcutaneous emphysema; any other pulmonary diseases impeding image acquisition (i.e., significant pleural effusion, previous pneumonectomy); and an inability to undergo LUS examination.

#### Outcomes

The study’s primary outcome was disease severity. We divided modalities into groups and graded them as follows: 0—no oxygen administration necessary; 1—0–10 L of oxygen nasally; 2—11–16 L of oxygen per bag mask; 3—HFNC (high-flow nasal cannula); and 4—MV (mechanical ventilation). A secondary outcome was fatality.

### 2.3. Data Extraction

Patient demographics, symptoms, laboratory tests, comorbidities, and treatment modalities were extracted from electronic medical records by the principal investigator (I.S.).

### 2.4. Acquisition Protocol

Lung ultrasound examinations were performed by two trained sonographers (I.S. and I.B.) using a Toshiba Nemio XG iSTYLE ultrasound system (Toshiba Medical Systems Corporation, Otawara, Japan) with a 1–6 MHz convex transducer. The transducer was in abdominal preset mode using a single focal point modality on the pleural line, a mechanical index starting from 0.7, depth from 10 to 15 cm, and gain was controlled to avoid over-saturation. The extent and severity of pulmonary infiltrations were described by a numerically repeatable LUS coefficient (Lung Ultrasound Score), proposed for COVID-19 pneumonia by Soldati et al. [26]. Fourteen areas (three posterior, two lateral, and two anterior for each lung) were examined completely intercostally to cover the widest possible area with a single scan. Changes were scored from 0 to 3, as presented in Table 1. For each patient, the stated scores in all fourteen zones were added together (ranging from 0 to 42) to obtain the total LUS score [26]. According to the part of the lung in which they were positioned, the 14 areas were grouped as apical, middle, and basal for further statistical analysis.

### 2.5. Bias

The LUS examinations were performed simultaneously by two team members who were blinded to the patients’ clinical data (I.S. and I.B.), reaching immediate consensus about the lung ultrasound score, and minimizing the information bias or measurement errors. Potential confounders of negative outcomes (e.g., arterial hypertension, cardiovascular diseases, or malignancy) were stated in all of the subgroups, and only two patients were excluded due to illnesses directly influencing the LUS score.

### 2.6. Study Size

This was a convenience sample of consecutive patients during the peak period of the pandemic in Croatia. The study size was calculated to accommodate a statistical power (beta error) of 80%, requiring a minimum of 130 patients. This coincided with the end of the pandemic peak and the occurrence of new viral strains.

### 2.7. Statistical Analysis

Categorical data were presented by absolute and relative frequencies. The normality of the distribution of continuous variables was tested by the Shapiro–Wilk test. Continuous data were described by the median and the limits of the interquartile range (IQR). The Mann–Whitney U test was used to compare the median between two groups and the Kruskal–Wallis test (post hoc Conover) was used to compare the median between two groups, whereas the Fisher’s exact test was used to analyze the differences between proportions. Logistic regression analysis (multivariate—stepwise method) was used to analyze independent factors associated with any respiratory support or a negative outcome. The receiver operating curve (ROC) was used to determine the optimal threshold, the area under the curve (AUC), specificity, and sensitivity of the tested parameters. All *p* values were two-sided. The level of significance was set at Alpha = 0.05. The statistical analysis was performed using MedCalc^®^ Statistical Software version 19.6 (MedCalc Software Ltd., Ostend, Belgium; https://www.medcalc.org; 2020, accessed on 20 April 2021.) and IBM SPSS Stat. 23 (IBM Corp. Released 2015. IBM SPSS Statistics for Windows, Version 23.0. Armonk, NY, USA).

### 2.8. Reporting

The study is reported in line with the STROBE reporting guideline for cohort studies; the STROBE checklist is available in Appendix A.

## 3. Results

### 3.1. Patients and Characteristics

From 1 November 2020, to 22 March 2021, we admitted 445 patients confirmed with COVID-19 by RT-PCR, and 133 patients were enrolled in the study (Figure 1). Two patients were excluded because of severe emphysema and chronic hypersensitivity pneumonitis.

The baseline characteristics of the included patients, including age, gender, day of illness, habits, comorbidities, and the respiratory support modality used, were comparable between the subgroups (Table 2).

The biochemical indicators upon presentation were similar no matter the highest respiratory modality used, except for spO2 (Table 3).

The severity of the ultrasonic parameters was also followed by the highest modality. The distribution of lung infiltrations was found to be lower in the upper regions of the lungs according to the LUS score (Friedman test, *p* < 0.001, Appendix A).

We have shown that the LUS score is only weakly correlated with clinical severity at admission to the hospital (Spearman’s rho 0.396, 95%CI 0.240 to 0.531, *p* < 0.001); however, the correlation is significantly greater with the progression in disease severity (Spearman’s rho 0.750, 95%CI 0.664 to 0.817, *p* < 0.001).

The LUS score had a predictive value in fatality (Kruskal–Wallis test, *p* < 0.001; Table 4). D-dimer levels were significantly higher in deceased patients (Kruskal–Wallis test, *p* = 0.002; Table 4).

### 3.2. Impact of LUS Score on the Highest Respiratory Support Modality Used

Two independent predictors yielded statistically significant contributions to the regression model: the LUS score and the age of the patient. The model accurately classified 91.6% of cases (Table 5, Appendix A).

Patients with higher LUS scores were more likely to need more respiratory support, either the high-flow nasal cannula (HFNC) with an LUS score cut-off value of >29 (Appendix A), or mechanical ventilation (MV) with a cut-off value of >30 (Table 5, Appendix A).

The LUS score and age of the patient predicted the need for mechanical ventilation, as shown in Figure 2 and Appendix A.

### 3.3. Impact of LUS Score and Risk Factors on the Patient Mortality

Two independent predictors yielded significant contributions to the regression model: LUS score and age of the patient. The model accurately classified 95.4% of cases (Table 6, Appendix A).

The LUS score and age of the patient were predictors of fatality, as shown in Figure 3 and Appendix A.

## 4. Discussion

LUS is very useful in the early diagnosis of COVID-19 pneumonia as a cheap, fast, radiation-free, and readily available method with sensitivity comparable to CT scans. This finding is even more important because obtaining computed tomography has proven to be a challenge in a pandemic setting [13,22].

Similar to earlier studies, the majority of patients in our study presented with bilateral interstitial changes in lung ultrasound examinations, such as B-lines, most frequently in posterior and basal zones of the lungs, where these changes were also of the highest intensity [15,23].

In previously published papers, LUS has been established as an excellent diagnostic technique in COVID 19 pneumonia, although there are limited data available concerning lung ultrasound as a prognostic tool for poor outcomes [14,15,16,17,18,19]. A study by Falgarone et al. even suggests that LUS has a better prognostic value for oxygen requirements than a chest CT scan, which is considered a gold standard for COVID-19 pneumonia [24].

The ability to predict the highest level of oxygen treatment necessary is valuable information, because it can help clinicians in planning patient care, potentially making decisions on transferring a patient to a more specialized intensive care center, or even to adapt the usual therapy protocols enabling a more individualized treatment approach. Some recent evidence suggests that high-dose corticosteroid therapy can help in treating severe COVID-19 pneumonia, and perhaps the indication of a poor prognosis in patients could be used to start high doses earlier in the course of the disease, thus preventing the worst outcomes [25,26].

Our study showed age as a predictor with a cut-off value of >68, both for mechanical ventilation and death. Previous studies have also shown age to be a predictor of mortality and ICU admission. A systematic review of 88 studies by Katzenschlager et al. revealed that patients who required ICU admission had a median age of 65, and patients who died had a median age of 71 [27].

According to our results, the LUS score showed a moderate correlation with oxygen requirements in patients at admission. We hypothesize that this discrepancy may be because interstitial changes, seen as B-lines, predate the later loss of aeration and progression to consolidation, as described in studies investigating ultrasound changes in ARDS [28,29]. Further studies are needed to explore the pathophysiology and development of ultrasound changes in COVID-19.

LUS was able to predict the need for HFNC or MV in COVID-19 patients. The cut-off values were similar, 29 and 30, respectively, suggesting that LUS cannot predict differentiations between these two groups. However, age was shown to only be predictive for MV in our population, which means that by combining it with a LUS score higher than 30, it can discriminate patients older than 68 who are more likely to need MV; those who are below that age who are more likely to need HFNC. To the best of our knowledge, this is the first study which has developed a model for predicting the need for HFNC in COVID-19 patients.

The AUC for LUS prediction of mortality in our study population was 0.87 at the cut-off value of LUS set to >30. The LUS score then yielded 81% specificity for mortality. Previous studies had slightly lower findings with AUC values of 0.72, 0.76, and 0.78, respectively, which could be attributed to fewer lung zones examined in the LUS protocols these earlier studies used, or the different patient populations [21,22,23].

As opposed to some of the earlier studies, we did not find any predictive value of laboratory blood markers, such as LDH, CRP or the lymphocytic count. However, in our population, D-dimer levels were found to be significantly higher among deceased patients.

It has to be noted that in our study, only 3.8% of patients enrolled were active smokers, which is a surprising finding because more than one-third (35%) of the Croatian population are active smokers, the third highest prevalence in the European Union [30]. Several studies have confirmed a very low prevalence of smokers among hospitalized COVID-19 patients [31,32,33]. It has been known for years that smokers are less likely to develop interstitial diseases such as sarcoidosis and hypersensitive pneumonitis, which is attributed to the possible protective effect of nicotine suppressing T-helper cell 1 immunity [34,35,36,37,38]. Further multi-center prospective studies are needed to confirm the potential protective role of smoking in COVID-19 patients.

### 4.1. Strengths

The median time from disease onset to LUS examination was 10 days, and the median time from hospital admission to LUS examinations was one day, which was the main strength of our study: this suggests that it was unlikely that many patients were at the peak of their illness at the time of LUS examination. At the time of examination, the median respiratory support administered was 2 L/min of oxygen flow through a nasal cannula.

We chose an LUS examination protocol with 14 lung regions suggested by Soldati et al., which differs slightly from 10 and 12 lung region protocols used in other previously mentioned studies [21]. A study by Hernandez et al. suggests that the 14-lung-region protocol is superior to other protocols [24]. Although this might tend to require a more extensive patient workload, we have used this advantageously and noticed two things: (i) COVID19 pneumonia tends to be more severe if apical regions are affected; and (ii) infiltration of the middle regions seem to correlate with the highest respiratory modality used, even better than the whole LUS score (Appendix A).

### 4.2. Limitations

Some limitations of our study should be highlighted. This study was performed in a single Clinical Medical Centre in Split, Croatia, with a relatively small number of patients. Not all patients admitted were included in this study due to the excessive workload and lack of resources and support staff.

Furthermore, we did not test interobserver variability because the two sonographers performed the examinations together due to time constraints as a result of the pandemic situation.

During this study, we noticed some limitations to the scoring system for the severity of pulmonary infiltrations proposed by Soldati et al., where consolidations and “white lung” were given the same score. Separating these two findings into different categories, as shown by Gutsche et al., should be considered, because consolidations are the most severe stage of COVID-19 pneumonia progression [18,39].

## 5. Conclusions

In our study, LUS at hospital admission was shown to be predictive of the need for HFNC, endotracheal intubation, and death. This finding can help in early risk stratification in COVID-19 patients, guiding further clinical and therapy decisions.

## Figures and Tables

**Figure 1 jpm-11-00757-f001:**
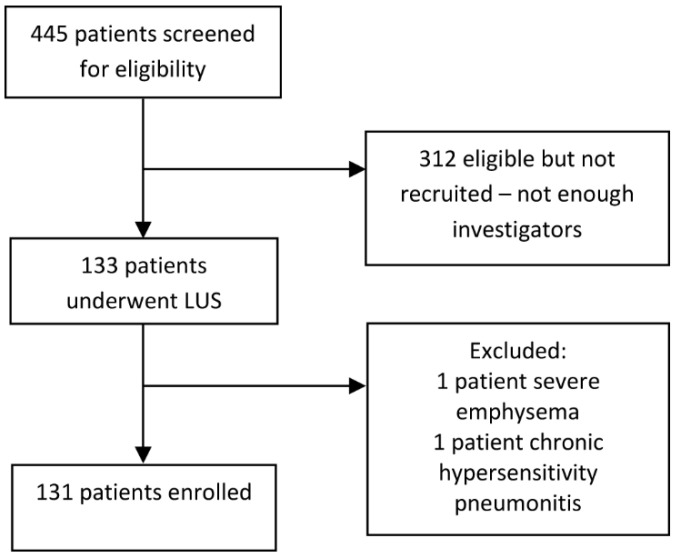
Flowchart of the included patients.

**Figure 2 jpm-11-00757-f002:**
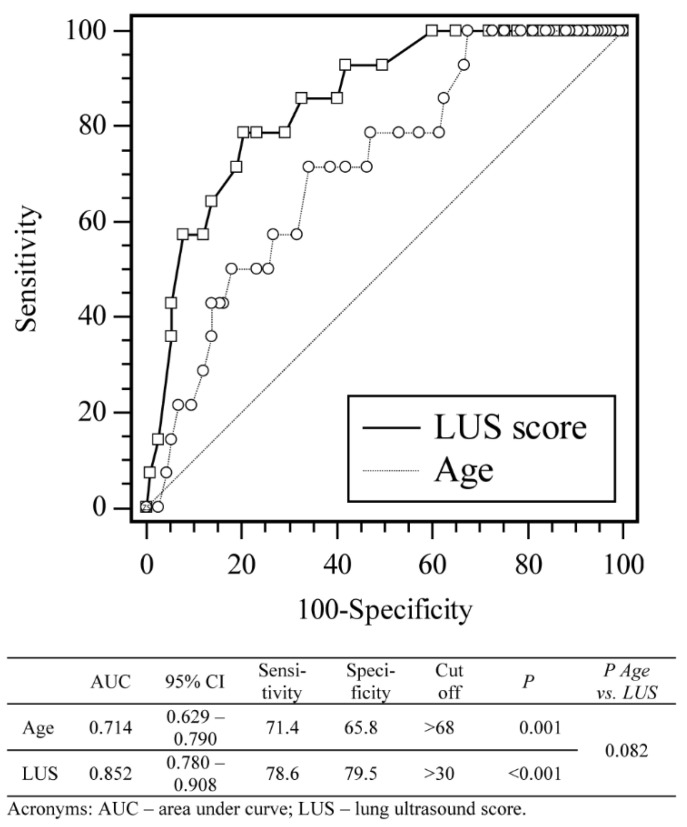
Receiver operating curve analysis of sensitivity, specificity, and cut-off values for LUS score and age according to highest respiratory modality (mechanical ventilation).

**Figure 3 jpm-11-00757-f003:**
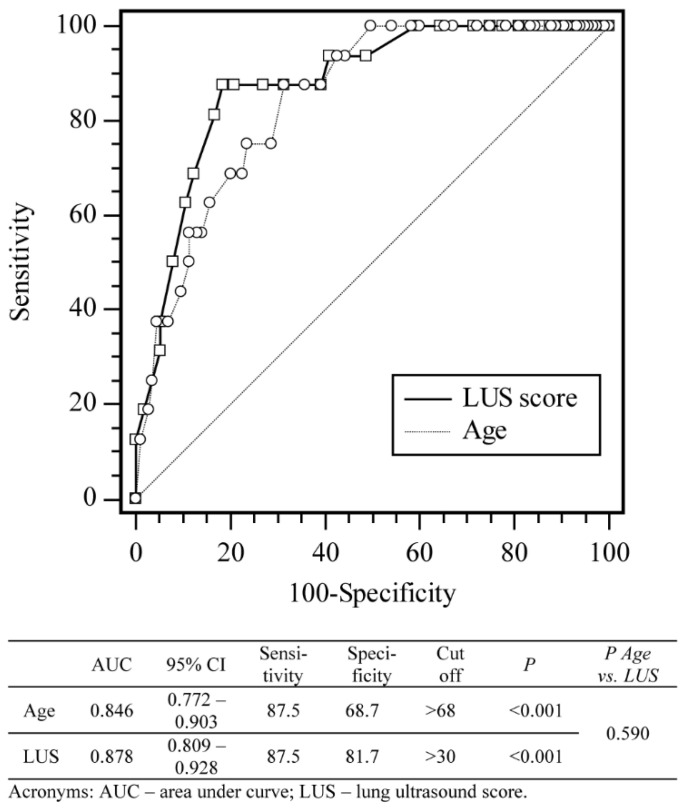
Receiver operating curve analysis of sensitivity, specificity, and cut-off values for LUS score and age according to the probability of a negative outcome (death).

**Table 1 jpm-11-00757-t001:** Lung ultrasound point scoring system.

Score	Ultrasonic Observation
0	Regular finding: existence of a regular and not thickened pleural line, with a sliding sign, and the presence of A-lines;
1	Some loss of aeration: irregular pleural line with some B lines;
2	Severe loss of aeration: broken pleural line; small-to-large consolidated areas with associated areas of white below the consolidated area;
3	Complete loss of aeration: scanned area shows large, dense consolidations; “white lung”.

**Table 2 jpm-11-00757-t002:** Baseline characteristics of patients according to comorbidity and respiratory support modality used.

Comorbidity	Number (%) of Patients						*p* *
	0 L	1–10 L	10–16 L or Venturi	HFNC	MV	Total	
Arterial hypertension	7 (50)	46 (65.7)	15 (75.0)	11 (73.3)	9 (69.2)	88 (66.7)	0.63
Diabetes	4 (28.6)	16 (22.9)	6 (30.0)	4 (26.7)	3 (23.1)	33 (25)	0.94
Cardiovascular disease	0 (0.0)	19 (27.1)	1 (5.0)	4 (33.3)	6 (46.2)	30 (22.9)	0.007
Liver failure	0 (0.0)	0 (0.0)	0 (0.0)	1 (6.7)	0 (0.0)	1 (0.8)	0.32
Lymphoma	2 (14.3)	2 (2.9)	0 (0.0)	1 (6.7)	0 (0.0)	5 (3.8)	0.20
Leukemia	0 (0.0)	0 (0.0)	1 (5.0)	0 (0.0)	2 (15.4)	3 (2.3)	0.03
Malignancy	3 (21.4)	7 (10)	1 (5.0)	3 (20)	1 (8.3)	15 (11.5)	0.44
Peripheral vascular disease	0 (0.0)	3 (4.3)	2 (10.0)	1 (6.7)	1 (8.3)	6 (4.6)	0.59
Dementia	0 (0.0)	0 (0.0)	0 (0.0)	1 (6.7)	0 (0.0)	1 (0.8)	0.32
Myocardial infarction	0 (0.0)	6 (8.6)	1 (5.0)	1 (6.7)	1 (8.3)	9 (6.9)	0.95
CVI or TIA	0 (0.0)	1 (1.4)	1 (5.0)	0 (0.0)	0 (0.0)	2 (1.5)	0.72
COPD	1 (7.1)	3 (4.3)	0 (0.0)	1 (6.7)	0 (0.0)	4 (3.1)	0.66
Rheumatological disease	1 (7.1)	1 (1.4)	1 (5.0)	0 (0.0)	0 (0.0)	3 (2.3)	0.39
Hemiplegia	0 (0.0)	0 (0.0)	0 (0.0)	0 (0.0)	1 (8.3)	1 (0.8)	0.09
Renal failure	0 (0.0)	3 (4.3)	1 (5.0)	0 (0.0)	0 (0.0)	4 (3.1)	>0.99

* Fisher’s Exact Test Abbreviations: COPD—chronic obstructive respiratory disease; CVI—cerebrovascular insult; HFNC—high-flow nasal cannula; LUS score—lung ultrasound score; MV—mechanical ventilation; TIA—transient ischemic attack.

**Table 3 jpm-11-00757-t003:** Biochemical parameters according to the highest level of respiratory support used.

	Mean ± SD or Median (IQR) According to the Highest Used Level of Respiratory Support Modality					*p* *
	0 L (*n* = 14)	1–10 L (*n* = 69)	11–16 L or Venturi (*n* = 20)	HFNC (*n* = 14)	MV (*n* = 14)	
LUS score	10 (3–21)	24 (19–26)	32 (27–35)	35 (32–38)	36 (31–38)	<0.0001 †
Age (years)	61.3 ± 16.0	63.8 ± 11.8	64.6 ± 10.4	64.6 ± 9.2	72.2 ± 8.7	0.115
CRP (mg/L)	118.8	74.5	85.6	121.0	125.6	0.340
Leukocyte count (10^9^/L)	7.49 ± 3.66	8.53 ± 3.54	8.22 ± 4.39	9.84 ± 5.97	8.04 ± 3.78	0.614
Neutrophils (%)	74.7 ± 13.7	79.4 ± 8.5	76.5 ± 10.1	82.6 ± 8.5	78.7 ± 8.4	0.193
Lymphocytes (%)	17.4 ± 12.0	14.1 ± 6.0	17.7 ± 9.9	12.1 ± 7.8	15.0 ± 9.6	0.201
D-dimer (µg/L)	0.76	0.92	0.85	0.98	1.89	0.390
LDH (U/L)	302	342	405	365	439	0.166
hs-Troponin (ng/L)	21.7	10.3	10.2	11.0	11.7	0.623
spO2 (%)	96.0	91.0	87.5	89.0	90.0	0.0003 ‡
pO2 (kPa)	10.00	7.18	6.92	6.89	5.89	0.069

* One-way ANOVA for normally distributed data (according to Kolmogorov–Smirnov test) or Kruskal–Wallis test (post hoc Conover test); † Pairwise comparison of subgroups detected significant differences in: 0 L vs. 1–10 L, 0 L vs. 10–16 L, 0 L vs. HFNC, 0 L vs. MV, 1–10 L vs. 10–16 L, 1–10 L vs. HFNC, 1–10 L vs. MV, and 11–16 L vs. HFNC. ‡ Pairwise comparison of subgroups detected significant differences in: 0 L vs. 1–10 L, 0 L vs. 10–16 L, 0 L vs. HFNC, 0 L vs. MV, and 1–10 L vs. 10–16 L.

**Table 4 jpm-11-00757-t004:** Differences in indicators according to mortality.

	Median (IQR)		Difference (95% CI)	*p* *
	Survived (*n* = 115)	Died (*n* = 16)		
Day of the illness	10.0 (8.0–13.0)	9.5 (6.0–12.0)	−1.0 (−4.0–1.0)	0.306
Presenting † respiratory support modality	1.0 (0.3–1.0)	1.0 (1.0–1.0)	0.0 (0.0–0.0)	0.375
Highest ‡ respiratory support modality	1.0 (1.0–2.0)	4.0 (3.0–4.0)	2.0 (2.0–3.0)	<0.001
LUS score	24.0 (19.5–29.0)	35.5 (32.0–38.0)	11.0 (7.0–14.0)	<0.001
CRP (mg/L)	82 (48–150)	117 (74–147)	−17.2 (−51.5–21.6)	0.372
Leukocyte count (10^9^/L)	7.6 (5.9–10.6)	7.7 (4.8–12.5)	0.0 (−2.5–2.2)	0.997
Neutrophils (%)	80.1 (74.6–85.3)	82.2 (74.2–89.0)	2.0 (−2.3–7.3)	0.276
Lymphocytes (%)	13.7 (9.0–18.4)	10.5 (6.3–18.5)	−2.9 (−6.9–0.9)	0.149
D-dimer (µg/L)	0.87 (0.60–1.51)	2.10 (1.53–3.37)	1.12 (0.55–1.80)	0.002
LDH (U/L)	361 (287–424)	439 (284–463)	40 (−33–118)	0.330
hs-Troponin (ng/L)	10.2 (7.2–18.4)	14.8 (9.8–50.9)	4.5 (−1.9–36.2)	0.171
spO2 (%)	91.0 (87.0–93.3)	89.5 (81.0–93.0)	−3.0 (−6.0–0.0)	0.093
pO2 (kPa)	7.14 (6.50–7.90)	6.64 (5.56–8.09)	−0.42 (−1.38–0.57)	0.419

* Mann–Whitney U test. † 0 = no support, 1 = 1–10 L, 2 = 11–16 L or Venturi; ‡ 0 = no support, 1 = 1–10 L, 2 = 11–16 L or Venturi, 3 = HFNC, 4 = MV. Abbreviations: HFNC—high-flow nasal cannula; IQR—interquartile range; LUS score—lung ultrasound score; MV—mechanical ventilation.

**Table 5 jpm-11-00757-t005:** Multivariate logistic regression analysis according to the highest respiratory support required (mechanical ventilation).

	ß	Wald	*p*	OR	95% CI
Age	0.081	5.051	0.025	1.08	1.01–1.16
LUS score	0.215	12.617	<0.001	1.24	1.10–1.40
Constant	−14.044	17.192	<0.001		

Acronyms: LUS score—lung ultrasound score.

**Table 6 jpm-11-00757-t006:** Multivariate logistic regression analysis according to mortality.

	ß	Wald	*p*	OR	95% CI
Age	0.232	13.510	<0.001	1.26	1.14–1.43
LUS score	0.344	14.589	<0.001	1.41	1.18–1.68
Constant	−28.864	18.882	<0.001		

Acronyms: LUS score—lung ultrasound score.

## Data Availability

The data presented in this study are available on request from the corresponding author.

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
