# Peer review of "Role of Lung Ultrasound in Predicting Clinical Severity and Fatality in COVID-19 Pneumonia"

_jpm, 2021, doi:10.3390/jpm11080757_

Round 1
Reviewer 1 Report
Dear authors,
The manuscript is interesting. The english should be reviewed by an english speaker especially the introduction there are some grammatical errors.
The manuscript is well written for the most part. Moreover, It should be detailed either in the introduction or the discussion the existing severity and mortality predictors of covid 19. Such as CT scan for example to see the difference with LUS for example.
In the results section, I think there are too many tables and readers may get lost, for example table 2. The authors should try to improve their results presentation so it is more easy to read and that the main results are highlighted.
Reviewer 2 Report
The manuscript investigates lung ultrasound findings in covid-19 patients at hospital admission in correlation to the disease severity during the clinical admission including mortality. Diagnostic accuracy of Lung ultrasound scores, were also compared with general infection parameter.
This study is of great importance for the clinical lung ultrasound community and for the clinical covid management.
The manuscript is well written with proper English and structured according to the authors guidelines. However, it has some limitations regarding reporting of methods and some statistical aspects as well as relevant literature, therefore it should be considered for publication with minor revision.
General comments:
Manuscript contains some formating issues such as "error reference not found" (page 4 line 146). In addition tables 2 is twice in the manuscript page 5.
Specific comments:
Title: Ok
Abstract: ok
page 1 line 27. I would recommend change “good” predictive power to something more specific like “high” or more pecific
Introduction:
page 1 line 40 to my knowledge adipositas-obesity is a high risk Factor for a severe covid-19 , and might be added.
The introduction of LUS page 2 line 49 - 52 could be improved getting appropriate literature describing the general use of lung ultrasound in its diverse applications.
In addition, current findings show the usability of lung ultrasound for COVID19 screening at hospital admissions, showing a high negative predictive value for LUS as found by Pivetta [1] or Gutsche et al. [2]. This information could be used to build the momentum of the introduction, investigating the correlation of LUS for estimation of disease severity. Or this important Information could be used for discussion of the findings such as on page 9 line 225-229.
Methods:
Were there other in- exclusion criteria regarding age or pregnancy status?
Acquisition protocol:
page 2 line 98 sonography settings could be described more detailed, regarding placement of focal zones, Harmonic imaging, used presets including safety indexes like mechanical index MI.
For easier reading, it could be considered to present the scores as a table.
important! Was the was derived in awareness of the patient’s outcome, the anamnesis ect.? Was it performed blinded- non blinded?
Results:
table 2.
In my opinion it should be considered to remove the difference between each parameter. There is no additional information and might confuse the reader.
I would recommend to present the score infiltration regarding outcome in a separated table, if needed. The exact test could be performed also for one outcome level over the regions, so the statistical significance could be presented below the table. In addition, in my opinion it's not makes much sense to present scores versus regions. Except that in all regions the score increases for fatal cases, no further information is extracted out of this information.
figure 2: very nice comparison of LUS score and age. Maybe to add a significant parameter such as D-dimer to the figure, demonstrating the strength of lung ultrasound over blood analysis.
Discussion: ok
The authors adapted the LUS Score system of soldati et al. 2020. It could be discussed if this score- feature classification is sufficient based on the examiners experience after the study. Or if it could be improved by proposed by other studies (Gutsche et al. [2]).
Limitations. OK
Bibliography
[1] |
E. Pivetta, A. Goffi, M. Tizzani, S. Locatelli, G. Porrino and I. Losano, "Lung Ultrasonography for the Diagnosis of SARS-CoV-2 Pneumonia in the Emergency Department," Ann. Emerg. Med., vol. 77, 2021. |
[2] |
H. Gutsche, T. Lesser, F. Wolfram and T. Doenst, "Significance of Lung Ultrasound in Patients with Suspected COVID-19 Infection at Hospital Admission," Diagnostics, vol. 11, no. 6, p. 921, 2021. |
Author Response
Point 1: Manuscript contains some formating issues such as "error reference not found" (page 4 line 146). In addition tables 2 is twice in the manuscript page 5.
Response 1: We have corrected these issues.
Point 2: page 1 line 27. I would recommend change “good” predictive power to something more specific like “high” or more pecific
Response 2: We agree and made the appropriate change.
Point 3: page 1 line 40 to my knowledge adipositas-obesity is a high risk Factor for a severe covid-19 , and might be added.
Response 3: We have added obesity as a risk factor, with a referenced study.
Point 4: The introduction of LUS page 2 line 49 - 52 could be improved getting appropriate literature describing the general use of lung ultrasound in its diverse applications. In addition, current findings show the usability of lung ultrasound for COVID19 screening at hospital admissions, showing a high negative predictive value for LUS as found by Pivetta [1] or Gutsche et al. [2]. This information could be used to build the momentum of the introduction, investigating the correlation of LUS for estimation of disease severity. Or this important Information could be used for discussion of the findings such as on page 9 line 225-229.
Response 4: We have added the section about LUS being comparable to X-ray and CT scans in certain pulmonary diseases. Further we have added the part about LUS usability in COVID19 screening at hospital admissions.
Point 5: Methods: Were there other in- exclusion criteria regarding age or pregnancy status?
Response 5: We did not exclude patients by age or pregnancy status. In our hospital, pregnant women with COVID19 were treated mostly at the appropriate COVID department in Gynaecology Clinic.
Point 6: Acquisition protocol:page 2 line 98 sonography settings could be described more detailed, regarding placement of focal zones, Harmonic imaging, used presets including safety indexes like mechanical index MI. For easier reading, it could be considered to present the scores as a table.
Response 6: As requested we have added details about transducer preset mode, focal point, mechanical index, depth and gain. Further, the table with LUS scores was added.
Point 7: important! Was the was derived in awareness of the patient’s outcome, the anamnesis ect.? Was it performed blinded- non blinded?
Response 7: Our sonographers were blinded to the clinical data of all patients, and accordingly this part was added in the Bias section.
Point 8: Results: table 2. In my opinion it should be considered to remove the difference between each parameter. There is no additional information and might confuse the reader.
Response 8: We have removed the differences from the Table 2 as suggested by the reviewer.
Point 9: I would recommend to present the score infiltration regarding outcome in a separated table, if needed. The exact test could be performed also for one outcome level over the regions, so the statistical significance could be presented below the table. In addition, in my opinion it's not makes much sense to present scores versus regions. Except that in all regions the score increases for fatal cases, no further information is extracted out of this information.
Response 9: We agree. The part with scores in each of the regions was removed from the tables, and presented in the existing supplementary materials/files.
Point 10: figure 2: very nice comparison of LUS score and age. Maybe to add a significant parameter such as D-dimer to the figure, demonstrating the strength of lung ultrasound over blood analysis.
Response 10: We kindly accept the suggestion and have presented this comparison in the newly added Supplementary file 4. However, we wanted to retain focus on the LUS score results on mortality and severity prediction.
Point 11: The authors adapted the LUS Score system of soldati et al. 2020. It could be discussed if this score- feature classification is sufficient based on the examiners experience after the study. Or if it could be improved by proposed by other studies (Gutsche et al. [2]).
Response 11: We have indeed noticed that the score suggested by Soldati et al. could be improved since consolidations and “white” lung changes are put in the same score. Accordingly, this discussion was added in the Limitations section.